# Do People with Intellectual Disabilities Have a Better Quality of Life If They Are Physically Active?

**DOI:** 10.3390/ejihpe15020014

**Published:** 2025-01-30

**Authors:** Carmen Ocete, Adolfo Rocuant-Urzúa, María Fernández-Rivas, Evelia Franco

**Affiliations:** 1GICAF Research Group, Education, Research Methods and Evaluation Department, Faculty of Human and Social Sciences, Universidad Pontificia Comillas, 28049 Madrid, Spain; mcocete@comillas.edu (C.O.); mfrivas@comillas.edu (M.F.-R.); 2Facultad de Educación, Campus San Joaquín, Pontificia Universidad Católica de Chile, Santiago 8320000, Chile; 3Department of Communication and Education, Faculty of Psychology and Education, Loyola Andalucía University, 41704 Seville, Spain; efranco@uloyola.es

**Keywords:** intellectual disability, sport, physical activity, logistic regression

## Abstract

In the context of the practice of physical sports activities (PSA), people with intellectual disabilities (PID) have up to a 62% lower possibility of responding to internationally agreed-upon physical activity requirements, showing a higher risk of presenting serious health problems. This study aimed to analyze the levels of perception towards the latent construct of quality of life that a PID who practices PSA would have with respect to those PIDs who do not practice PSA. The sample consisted of 371 PIDs, of whom 278 practiced PSA and 93 did not practice PSA. The instrument used for data collection was the INICO-FEAPS scale. Methodologically, 12 binary logistic regression models were fitted. The results identify greater possibilities of perception by PIDs who perform DFA at the level of self-determination (4.88 OR), rights (5.0 OR), social inclusion (2.06 OR), personal development (2.80 OR), interpersonal relationship (3.19 OR), material well-being (4.41 OR) and quality of life (3.97 OR). Furthermore, when grouping the dimensions by independence (3.67 OR), social (4.34 OR) and well-being (3.91 OR), the odds were favourable toward those PIDs performing PSA. In conclusion, PIDs who perform PSA may present greater possibilities of having the quality of life construct.

## 1. Introduction

International legislation, through the Convention on the Rights of Persons with Disabilities (PD) ([27]), has globally promoted the protection and achievement of equality in all human rights and fundamental freedoms of persons with disabilities, exposing in Article 30 the importance of access and participation by persons with disabilities to culture, leisure and sports as key elements for equality. International efforts are focused on promoting the practice of physical activities in people with disabilities (PD) ([19]; [33]) due to the benefits obtained physically, emotionally, psychologically and socially ([33]), which in the specific case of PD becomes more relevant if possible, highlighting the impact obtained at the social level ([10]). Along these lines, specifically among people with intellectual disabilities (PID), there is evidence of positive effects caused by the practice of physical activities ([6]; [20]; [16]) as a determinant to demonstrate empirical findings regarding quality of life ([10]). This is understood as the perception of personal well-being linked to the achievement of basic human requirements and relevant aspects in people’s lives influenced by personal and environmental factors ([30]).

In contrast to the above, social policies related to the access of PDs to general contexts have historically been subject to continuous restrictions in accessing services that promote their social inclusion ([18]), particularly access to practicing physical sports activities ([4]; [10]; [14]; [15]). Moreover, in the context of physical activity, recreation and sports participation, PDs were up to 62% less likely to respond to internationally agreed-upon physical activity requirements, showing higher risks of presenting severe levels of health ([20]). Young people experiencing disability face many obstacles when it comes to taking part in physical activities, such as a lack of sports programmes, lack of experience in sports ([11]) or other problems of a physical, social or emotional nature (e.g., overprotective adults, social isolation, significant time in treatment programmes and difficulties in travelling to training sessions and games) ([26]). These barriers lead to low levels of physical activity among this group, and the need to create more accessible practice environments becomes an ongoing necessity ([21]). In Spain, according to official data from 2018, a total of 274,833 PIDs were recognized with disability certificates with a degree equal to or greater than 33%. In this country, the General Law on the Rights of Persons with Disabilities and their Social Inclusion ([5]) emphasizes through chapter seven the need to have different social services that attend to improving the quality of life of this population group. In this sense, sports are identified as a fundamental service for the development of personality, inclusion in society and improvement of well-being ([10]).

There are several challenges of measuring quality of life in PIDs, including conceptual, methodological, cultural and communication issues ([3]; [28]). On one hand, [24] ([24]) highlight the importance of considering multiple domains, such as autonomy, social inclusion and well-being, and they point out the difficulties in developing standardized, universally applicable measures. The need to consider this multidimensional nature of quality of life while emphasizing the challenge of capturing subjective experiences has also been explored by [9] ([9]). Interestingly, cultural context has been found to be a conditioning factor for the perception of quality of life among PIDs, and it is widely accepted that assessment tools should be adapted to reflect cultural values and norms ([28]).

The international literature presents empirical evidence regarding PID who practice physical activities and/or sports, which show positive results in different dimensions of quality of life ([7]; [10]; [22]; [2]). In this sense, positive associations between the practice of physical sports activities and self-determination ([10]; [22]), social inclusion ([2]; [10]) and physical, emotional and material well-being ([7]; [10]) are reported. Studies such as [10] ([10]) report statistically significant results in favor of PIDs who practice physical sports activities in contrast to those who do not engage in physical sports activities. With a sample of 380 PIDs, they identified significant effects of the practice of physical sports activities on self-determination, emotional well-being, and social inclusion, as well as independence socialization, and well-being. However, in the case of PIDs who engaged in physical sports activities, significant differences were identified in favor of those who did it on a non-regular basis with respect to those who practiced it regularly in the dimensions of emotional well-being, social inclusion and personal development.

Based on the above, [24] ([24], [25]) proposed to address this issue through the design of a multidimensional tool for quality of life that considered the following elements: self-determination, rights, emotional well-being, social inclusion, personal development, interpersonal relationships, material well-being and physical well-being. The authors state that these dimensions belong to the realms of independence, socialization and well-being ([31]).

Considering the above, the aim of this study is to analyze the possibilities of perception towards the construct of quality of life that a PID who practices physical sports activities would have in Spain with respect to PIDs who do not practice physical sports activities. Thus, this investigation aims to answer the research question ‘Do people with intellectual disabilities have a better quality of life if they are physically active?’. To answer this, it is hypothesized that PIDs who practice physical sports activities in Spain have greater possibilities of having a higher perception of quality of life than PIDs who do not practice physical sports activities.

## 2. Materials and Methods

### 2.1. Design and Participants

To fulfill the purpose of this research, a cross-sectional predictive design was considered.

The selection was selected non-probabilistically ([12]) with convenience sampling ([8]). The identity of the participants was anonymized, and, before considering participation, the purposes of the study were explained. Participation was voluntary and required the signing of an informed consent form. In this sense, this research adheres to the Declaration of Helsinki on research integrity and has the approval of the Ethics Committee of the Pontificia Universidad Comillas (N. 2022/46). Finally, this study considered a sample of N = 371 PIDs from Spain (see Table 1), composed of men (n = 204) and women (n = 167) between 13 and 65 years of age (M = 28.3; SD = 12.6). The PIDs in the study were grouped according to those less than or equal to 18 years of age (21%) and those over 18 years of age (79%). The data distinguish between PIDs who did not practice sports (n = 93) and those who did practice sports (n = 278).

### 2.2. Instrument

The collection instrument was applied between November 2022 and April 2023.

Quality of life: To assess quality of life, the INICO-FEAPS Scale, a comprehensive assessment of the quality of life of people with intellectual or developmental disabilities ([29]), was used. This instrument considers 72 items distributed across 8 dimensions of interest: self-determination (e.g., I participate in the decisions made in my home), rights (e.g., I am allowed to participate in the design of my individual plan), emotional well-being (e.g., I feel self-confident), social inclusion (e.g., I have friends who do not have a disability), personal development (e.g., I am taught things that interest me), interpersonal relationships (e.g., I express my emotions and feelings in front of my friends), material well-being (e.g., I have the services and supports I need) and physical well-being (e.g., I engage in sports or leisure activities). The response levels were measured with a four-level Likert-type scale (1–4). PIDs who responded with options 3 and 4 (frequently and always) were considered to present a higher perception towards each dimension of quality of life proposed by [24] ([24], [25]), while PIDs who responded with options 1 and 2 (never and sometimes) were considered to present a lower perception towards the construct. Internal consistency was assessed using Cronbach’s alpha coefficient to consider the scores associated with each dimension, reporting values between α = 0.59 (physical well-being) and α = 0.91 (quality of life) (see Table 2).

For the purposes of this study, the means of the scores obtained were used (see Table 3) and dichotomized into two categories. PIDs with an average score below the mean were categorized as people with a lower level of perception of the construct, and those with an average score equal to or above the mean were categorized as people with a higher perception of self-determination.

Practice of physical sports activities: The participants were asked whether they engaged in physical sports activities in their free time, a question to which they could answer yes or no.

### 2.3. Statistical Analysis

The statistical analyses associated with this research were performed using R software, with RStudio 2024.12.0+467 version ([23]) utilizing descriptive (see Table 1 and Table 2), descriptive and correlational (see Table 3), and independence analyses by means of Chi-square testing ([1]) (see Table 4). Finally, 12 binomial logistic regression models were fitted ([13]). In addition, the goodness-of-fit of the models was analyzed by the Receiver Operating Characteristic (ROC). ROC represents the proportion of true positives versus the proportion of false positives of the model. The latter allowed us to explore to what extent the independent variable (sports practice) would enable a higher perception of each of the dimensions of interest of this study regarding quality of life (dependent variables).

Based on the above, the equation associated with each of the adjusted logistic models can be expressed as follows:$$\log\left(\frac{Pr\left(Variable=1\right)}{1-Pr\;\left(Variable=1\right)\;}\right)=\beta_{0}+\beta_{1}Sports\;practice+\varepsilon$$
where $\log\left(\frac{Pr\left(Variable=1\right)}{1-Pr\left(Variable=1\right)}\right)$ is the possibility of having a higher perception of the quality of life dimension; $\beta_{1}$ is the practice of physical sports activities.

## 3. Results

The descriptive analysis of this study (see Table 2) allows us to categorize people with intellectual disabilities (PID) (N = 371) according to their level of perception towards the different dimensions of quality of life utilized in this study. As indicated in the section on instruments, participants with scores below the mean in a dimension were considered to show relatively low values of perception, while those with scores above the mean were categorized as participants with relatively high values of perception in a dimension. These categories allowed us to identify a symmetrical percentage distribution for the variables of rights, emotional well-being, social inclusion and quality of life, as well as a less symmetrical percentage distribution for the variables of self-determination, personal development, interpersonal relationships, physical well-being and material well-being. When grouping the results according to dimensions, it is possible to identify symmetrical percentage distributions in the results.

Table 3 presents the descriptive and correlational results. In this sense, it is possible to identify positive and statistically significant associations between all the dependent variables of the study and the independent variable, with correlation values between r = 0.13 and r = 0.38. As for the type of association between sports practice and the independence dimension (r = 0.34, *p* < 0.01), the social dimension (r = 0.36, *p* < 0.01) and the well-being dimension (r = 0.25, *p* < 0.01), the evidence shows linear and statistically significant associations between the variables (see Table 3).

Table 4 presents the results associated with the analyses of independence for the different variables of interest in the study. In this sense, the results suggest the existence of a direct and statistically significant relationship between the variable of sports practice and most of the dependent variables of the study. However, the analysis of independence between the variables of sports practice and emotional well-being presented a level of statistical significance slightly above the expected value (χ^2^ (1, n = 371) = 3.794, *p* = 0.051). Additionally, the analyses of independence between the independent variable and the dimensions of interest suggest a direct and statistically significant relationship.

The results derived from the logistic regression model fit (see Table 5) show that those PIDs who practiced sports were more likely to have a higher perception of quality of life than their peers who did not practice sports in most of the dimensions of interest. Specifically, the model fit shows that the group of PIDs who practiced sports increased by 5 odds ratios (OR) in the rights dimension and 2.069 OR in the social inclusion dimension. When reviewing the results grouped by dimension, the independence dimension reported an increase of 3.67 OR in PIDs who practiced sports with respect to their peers who did not practice sports. The social dimension showed an increase of 4.34 OR in PIDs who practiced sports with respect to their peers who did not practice sports, and the well-being dimension showed an increase of 3.91 OR in the odds of having a higher perception of the construct with respect to their peers who did not practice sports.

The results associated with the goodness of fit of the models (see Table 5) report a fit of the model associated with the self-determination variable with a ROC of 64%, the fit of the model associated with the rights variable with a ROC of 64%, the fit of the model associated with the interpersonal relationship variable with a ROC of 61%, the fit of the model associated with the interpersonal relationships variable with a ROC of 61%, the fit of the model associated with the self-determination variable with a ROC of 64%, the model fit associated with the material well-being variable with a ROC of 62%, a model fit associated with the quality of life variable with a ROC of 62% and the fits associated with the independence, social and well-being dimensions with a ROC of 61.4%, 63.4% and 62.1%, respectively.

## 4. Discussion

The aim of this study was to analyze the possibilities of perception towards the construct of quality of life that PIDs in Spain who practice physical sports activities would have with respect to PIDs who do not practice physical sports activities. Thus, this investigation aimed to answer the research question ‘Do people with intellectual disabilities have a better quality of life if they are physically active?’. Towards this aim, it was hypothesized that PIDs who practice physical sports activities in Spain have a higher perception of quality of life than those PIDs who do not practice physical sports activities.

Using a logistic regression model, statistically significant differences were identified in favor of the PIDs who practice physical sports activities. Specifically, this group had greater perception towards the dimensions of self-determination, rights, social inclusion, personal development, interpersonal relationships and material well-being towards the global construct of quality of life.

When comparing the results with related studies in this field, it is possible to identify similarities. In this sense, the likelihood of having a higher perception towards the dimension of self-determination increases when PIDs practice physical sports activities; these results contribute to the existing evidence associated with a positive tendency in favor of PIDs who practice physical sports activities ([10]; [22]). The findings associated with the dimension of social inclusion identify greater possibilities on the part of PIDs who practice physical sports activities; these results are consistent with those reported by different studies ([2]; [10]). The material well-being dimension results reinforce the findings of [7] ([7]) and [10] ([10]), as PIDs in this experiment were also more likely to have a higher level of perception towards material well-being. Finally, it is possible to identify results that reinforce the findings of [10] ([10]) regarding the overall level of perception towards the quality of life construct in favor of PIDs who engage in physical sports activities. Specifically, this study identifies a higher perception of quality of life on the part of PIDs who engage in physical activities and sports.

Regarding the differences with other studies of interest, this research reports higher perceptions of the dimension of interpersonal relationships. This differs from the results presented by [10] ([10]), who found no statistically significant differences between PIDs who perform physical sports activities and those PIDs who do not perform physical sports activities. Similarly, in other experiments the results associated with PIDs who perform physical sports activities in the physical and emotional well-being dimensions ([7]; [10]) differ from the findings of this research. Although the results of this study show that in the rights dimension, PIDs who practice physical sports activities have greater possibilities of perception towards this dimension and in the dimension of personal development, this group would present more possibilities than their peers who do not perform physical sports activities. However, there are no studies that allow us to contrast these results. Based on these differences, it would be important to analyze the different components derived from each dimension, since there could be elements that have not been addressed by the literature.

The results of this research have the potential to contribute significantly to the diffusion, creation and evaluation of physical sports activity programs aimed at improving the quality of life of PIDs. In terms of dissemination, having empirical evidence that relates sports practice with a greater perception of quality of life will allow families, caregivers and communities to promote the participation of this population in sports activities. Regarding the creation of sports programs, the findings of this study underscore the importance of incorporating specific dimensions of quality of life that strengthen the technical-tactical skills associated with each sports discipline. Finally, in the field of active sports program evaluation, the results of this research offer a framework for analyzing and considering the various dimensions of quality-of-life constructs. This will allow adjusting and optimizing planned actions, enriching the sports experience of PIDs and enhancing the positive impact of these initiatives.

### 4.1. Limitations

Based on the above, it is essential to reflect on limitations and future lines of interest. Regarding the limitations, it is important to highlight those methodological decisions linked to the selection of participants, since they were selected according to access possibilities and in a non-probabilistic manner; therefore, it is not possible to generalize the results to the entire population of PIDs. A second element, not addressed, considers that the physical sports activities in which the PIDs participate could present substantive differences with respect to their objectives and planning around the construct of quality of life and the possible emphasis on some of its dimensions. Finally, this study did not consider elements linked to interpretative analysis from the PIDs.

With respect to future studies, both academics and stakeholders should delve deeper into the gaps associated with the findings of this research and the results of other studies. This could include qualitative studies to enrich and complement the interpretations regarding elements derived from the dimensions of interest, such as social relationships (interpersonal relationships); rest and access to health care (physical well-being); life satisfaction; self-concept and absence of interest or negative feelings (emotional well-being); privacy and confidentiality (law) and training; and learning and competence at work (personal development).

### 4.2. Recommendations for Future Research

Future lines of research should consider adjustments of statistical models that incorporate variables of interest regarding barriers and facilitators associated with the practice of physical sports activities by PIDs. With respect to barriers, it is possible to consider the training of teachers, programming and the time of physical activities ([17]). In addition, it would be possible to incorporate into the model other elements identified as barriers to the promotion of physical activity, such as the personal characteristics of PIDs, family members, social elements, and economic and environmental variables ([14]). In a complementary manner, it will be important to reflect and consider in future studies the incorporation of variables related to the socioeconomic level of PIDs, family composition, the level of education of parents and/or caregivers, the number of weekly hours of physical sports activity practice and the expectations of parents and/or caregivers. These are perceived as barriers to the promotion and participation of PIDs in regular physical sports activities ([14]). Faced with this, this group should receive more information about the benefits derived from the practice of physical sports activities ([15]) and its influence towards greater autonomy on the part of the PID. At the level of facilitators, future studies should integrate analyses of the level of peer acceptance and the support of other teachers or interested specialists ([32]), as well as the expectations on the part of those who lead physical activity projects for this population group ([14]).

At the methodological level, it is important to incorporate experimental or quasi-experimental studies that use pre-tests and post-tests to find the results of the practice of physical sports activities on the quality of life of PIDs through quality-of-life measurements. In addition, it is relevant to complement the analyses related to the level of understanding of the items associated with the data collection scale used to determine the levels of quality of life perceived by the PIDs with validity analyses associated with the response processes ([8]). This would enrich the uses and interpretations of the scores associated with the latent of interest.

## 5. Conclusions

The findings above show that people with intellectual disabilities (PIDs) who take part in physical sports are more likely to have a positive view of various aspects of their quality of life—such as independence, social connections and overall well-being—compared to those who do not engage in sports. This supports earlier research by [7] ([7]), which found a positive link between physical sports and improvements in independence, social life, and well-being.

Based on the evidence from this study, it seems likely that PIDs who participate in sports have a better perception of their quality of life than those who do not. Specifically, they tend to have a more positive outlook on areas like self-determination, rights, social inclusion, personal development, relationships, material well-being and their overall quality of life (see Table 5).

This research supports the idea that encouraging physical activity for people with intellectual disabilities should be a priority in public policies and international agreements, as highlighted by [15] ([15]).

## Figures and Tables

**Table 1 ejihpe-15-00014-t001:** Characteristics of the participants.

Variables	n	%
**Gender**		
Men	204	55
Women	167	45
**Age**		
Less than or equal to 18	78	21
Over 18	293	79
**Sports Practice**		
No	93	25
Yes	278	75
**N**	**371**	

**Table 2 ejihpe-15-00014-t002:** Distribution of dependent variables according to level of perception and Cronbach’s Alpha.

Variable	n	%	Cronbach’s Alpha
**Self-determination**			0.79
Lower perception	172	46.4	
Higher perception	199	53.6	
**Rights**			0.68
Lower perception	185	49.9	
Higher perception	186	50.1	
**Emotional well-being**			0.65
Lower perception	181	49	
Higher perception	190	51	
**Social inclusion**			0.48
Lower perception	190	51	
Higher perception	181	48	
**Personal development**			0.75
Lower perception	200	53.9	
Higher perception	171	46.1	
**Interpersonal relationships**			0.65
Lower perception	169	45.6	
Higher perception	202	54.4	
**Material well-being**			0.64
Lower perception	222	59.8	
Higher perception	149	40.2	
**Physical well-being**			0.59
Lower perception	166	45	
Higher perception	205	55	
**Quality of life**			0.91
Lower perception	181	49	
Higher perception	190	51	
**Dimensions**			
**Independence**			0.86
Lower perception	191	51	
Higher perception	180	49	
**Social**			0.80
Lower perception	175	47	
Higher perception	196	53	
**Well-being**			0.76
Lower perception	182	49	
Higher perception	189	51	
**N**			
	**371**		

**Table 3 ejihpe-15-00014-t003:** Matrix of correlations between study variables, mean and standard deviation.

N°	Variable	1	2	3	4	5	6	7	8	9	10
1	Sports practice	1	0.37 **	0.38 **	0.13 **	0.13 *	0.25 **	0.33 **	0.23 **	0.19 **	0.37 **
2	Self-determination		1	0.71 **	0.22 **	0.32 **	0.70 **	0.60 **	0.53 **	0.21 **	0.79 **
3	Rights			1	0.26 **	0.36 **	0.69 **	0.58 **	0.61 **	0.26 **	0.81 **
4	Emotional well-being				1	0.49 **	0.33 **	0.37 **	0.30 **	0.45 **	0.56 **
5	Social inclusion					1	0.39 **	0.48 **	0.32 **	0.36 **	0.62 **
6	Personal development						1	0.62 **	0.71 **	0.31 **	0.86 **
7	Interpersonal relationships							1	0.48 **	0.24 **	0.78 **
8	Material well-being								1	0.31 **	0.76 **
9	Physical well-being									1	0.50 **
10	Quality of life										1
M (SD)			2.66	2.89	3.15	3.05	2.82	2.78	3.08	3.4	2.98
			(0.66)	(0.58)	(0.45)	(0.43)	(0.6)	(0.55)	(0.52)	(0.41)	(0.38)
**N°**	**Dimensiones**	**1**	**2**	**3**	**4**						
1	Sports practice	1	0.34 **	0.36 **	0.25 **						
2	Independence		1	0.76 **	0.57 **						
3	Social			1	0.63 **						
4	Well-being				1						
M (SD)			2.74	2.91	3.21						
			(0.58)	(0.42)	(0.34)						

Note: N = 371; ** *p* < 0.01, * *p* < 0.05.; M = mean; SD = standard deviation.

**Table 4 ejihpe-15-00014-t004:** Contingency matrix distribution of the variable sports practice as a function of quality of life.

Independent Variable	Category	Dependent Variable		Stadistical Test
**Sports practice**		**Self-Determination**		
		**Lower Perception**	**Higher Perception**	**χ^2^**
	No	69 (43.11)	24 (49.88)	
	Yes	103 (128.88)	175 (149.11)	
				37.182 ***
		**Rights**		
		**Lower Perception**	**Higher Perception**	**χ^2^**
	No	72 (46.37)	21 (46.62)	
	Yes	113 (138.62)	165 (139.37)	
				36.235 ***
		**Emotional Well-Being**		
		**Lower Perception**	**Higher Perception**	**χ^2^**
	No	54 (45.37)	39 (47.62)	
	Yes	127 (135.62)	151 (142.37)	
				3.79
		**Social Inclusion**		
		**Lower Perception**	**Higher Perception**	**χ^2^**
	No	60 (47.62)	33 (45.37)	
	Yes	130 (142.37)	148 (135.62)	
				8.09 **
		**Personal Development**		
		**Lower Perception**	**Higher Perception**	**χ^2^**
	No	67 (50.13)	26 (42.86)	
	Yes	133 (149.86)	145 (128.13)	
				15.46 ***
		**Interpersonal Relationships**		
		**Lower Perception**	**Higher Perception**	**χ^2^**
	No	62 (42.36)	31 (50.63)	
	Yes	107 (126.63)	171 (151.36)	
				21.187 ***
		**Material Well-Being**		
		**Lower Perception**	**Higher Perception**	**χ^2^**
	No	77 (55.64)	16 (37.35)	
	Yes	145 (166.35)	133 (111.64)	
				25.959 ***
		**Physical Well-Being**		
		**Lower Perception**	**Higher Perception**	**χ^2^**
	No	55 (41.61)	38 (51.38)	
	Yes	111 (124.38)	167 (153.61)	
				9.64 **
		**Quality of Life**		
		**Lower Perception**	**Higher Perception**	**χ^2^**
	No	68 (45.37)	25 (47.62)	
	Yes	113 (135.62)	165 (142.37)	
				28.122 ***
		**Independence**		
		**Lower Perception**	**Higher Perception**	**χ^2^**
	No	69 (48)	24 (45)	
	Yes	122 (143)	156 (135)	
				24.43 ***
		**Social**		
		**Lower Perception**	**Higher Perception**	**χ^2^**
	No	68 (44)	25 (49)	
	Yes	107 (131)	171 (147)	
				32.159 ***
		**Well-being**		
		**Lower Perception**	**Higher Perception**	
	No	68 (46)	25 (47)	
	Yes	114 (136)	164 (142)	
				27.482 ***

Note: N = 371; ** *p* < 0.01, *** *p* < 0.001; () = expected values.

**Table 5 ejihpe-15-00014-t005:** Logistic regression coefficients of quality of life variables on the variable sports.

Fitted Models	Coefficients		OR	ROC
	$\beta_{0}$ ()	$\beta_{1}$ ()		
1. Self-determination	−1.06 (0.24) ***	1.59 (0.27) ***	4.884	64%
2. Rights	−1.23 (0.25) ***	1.61 (0.28) ***	5.00	63.81%
3. Emotional well-being	−0.033 (0.21)	0.50 (0.24) *	1.646	55%
4. Social inclusion	−0.60 (0.22) **	0.73 (0.25) **	2.069	57%
5. Personal development	−0.95 (0.23) ***	1.03 (0.26) ***	2.809	59%
6. Interpersonal relationships	−0.69 (0.23) **	1.16 (0.25) ***	3.196	61%
7. Material well-being	−1.57 (0.27) ***	1.48 (0.30) ***	4.414	62%
8. Physical well-being	−0.37 (0.21)	0.78 (0.24) **	2.177	57.3%
9. Quality of life	−1.00 (0.23) ***	1.38 (0.26) ***	3.971	62.2%
Dimensions	$\beta_{0}$ **()**	$\beta_{1}$ **()**	**OR**	**ROC**
10. Independence	−1.06 (0.24)	1.30 (0.27) ***	3.67	61.4%
11. Social	−1.00 (0.23)	1.47 (0.26) ***	4.34	63.4%
12. Well-being	−1.00 (0.23)	1.36 (0.26) ***	3.91	62.1%

Note: $\beta_{0}$ = intercept, $\beta_{1}$ = slope; OR = Odds ratio; () = Standard Error; * *p* < 0.05, ** *p* < 0.01; *** *p* < 0.001; Receiver Operating Characteristic (ROC); N = 371.

## Data Availability

The data supporting the results of this study can be requested from the corresponding author.

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
