# Peer review of "Do People with Intellectual Disabilities Have a Better Quality of Life If They Are Physically Active?"

_ejihpe, 2025, doi:10.3390/ejihpe15020014_

Round 1
Reviewer 1 Report
Comments and Suggestions for Authors
Thank you for the opportunity to review this paper. It is a well-conducted study with strong potential to contribute to future research. However, the paper is not ready for publication in its current form.
The study includes a solid sample size, which is a strength. The main concern is with the grammar and sentence structure, which make the paper difficult to follow. I strongly recommend the authors work with someone proficient in English to improve the clarity and readability of the manuscript. It would be unfortunate for this study not to be published because of these issues.
Here are a few specific suggestions:
- Methods Section: There seems to be a typo regarding the age range of participants (3 to 65 years). It seems unlikely that a 3-year-old could complete the type of questionnaire described. Please verify and correct this.
- Results Section: The results are repetitive, as much of the information in the text is already presented in the tables. I recommend keeping the text brief and focusing on key findings, with the tables providing detailed data.
- Discussion Section: The discussion would benefit from more depth. Instead of only comparing your study to others, explain the significance of your findings. What new insights do they provide? How can they move the field forward?
Overall, this is a strong paper, but the grammar and sentence structure made it difficult to read. I appreciate the effort the authors put into this study and hope these suggestions are helpful in improving the manuscript.
Comments on the Quality of English Language
The main issue with the paper is the quality of the English. The grammar and sentence structure make the manuscript difficult to follow, but this is an easy fix. I believe the authors deserve an opportunity to work with someone proficient in English to improve the clarity and readability of their work. Addressing this will ensure the study’s quality is fully appreciated.
Author Response
We would like to express our gratitude to the anonymous reviewers for providing us with such valuable feedback and comments regarding the submission to European Journal of Investigation in Health, Psychology and Education entitled "Do people with intellectual disability have a better quality of life if they are physically active?”. We do appreciate reviewers’ comments and suggestions, which we have attempted to address. It is our thought that the manuscript has been significantly improved, therefore.
We have highlighted the changes in the revised version of the manuscript in a yellow-coloured text aiming to facilitate reviewers’ evaluation.
Comments 1: Thank you for the opportunity to review this paper. It is a well-conducted study with strong potential to contribute to future research. However, the paper is not ready for publication in its current form.
The study includes a solid sample size, which is a strength. The main concern is with the grammar and sentence structure, which make the paper difficult to follow. I strongly recommend the authors work with someone proficient in English to improve the clarity and readability of the manuscript. It would be unfortunate for this study not to be published because of these issues.
Response 1: Thank you very much for your feedback in this regard. If the paper is finally accepted for publication in EJIHPE, we will submit the manuscript to a professional proofreading service to revise the English before the paper publication.
Here are a few specific suggestions:
Comment 2: Methods Section: There seems to be a typo regarding the age range of participants (3 to 65 years). It seems unlikely that a 3-year-old could complete the type of questionnaire described. Please verify and correct this.
Response 2: Thank you for your attention. It was certainly a mistake and it has been corrected (the age range was 13-65). (Line 117)
Comment 3: Results Section: The results are repetitive, as much of the information in the text is already presented in the tables. I recommend keeping the text brief and focusing on key findings, with the tables providing detailed data.
Response 3: Thank you very much for your comment. The manuscript has been revised and only key findings are now described in text. (Lines 168-177, 181-187, 190-197 and 201-212)
Comment 4: Discussion Section: The discussion would benefit from more depth. Instead of only comparing your study to others, explain the significance of your findings. What new insights do they provide? How can they move the field forward?
Response 4: Thank you very much for your comment. The manuscript has been revised and adjusted (lines 271 –282).
Comment 5: Overall, this is a strong paper, but the grammar and sentence structure made it difficult to read. I appreciate the effort the authors put into this study and hope these suggestions are helpful in improving the manuscript.
Response 5: Thank you very much for both your positive feedback and your suggestions to improve our manuscript.
Reviewer 2 Report
Comments and Suggestions for Authors
The manuscript is well-structured and provides an overview of the context and existing research on the relationship between physical activity and quality of life among people with intellectual disabilities. However, several important aspects are missing that would be essential for a comprehensive understanding of the subject:
1. The authors should include in the introduction a deeper exploration of the barriers faced by PIDs in accessing physical sports activities, as well as potential strategies to overcome these barriers. This would provide valuable context for interpreting the findings.
2. The introduction would benefit from a discussion on the specific challenges of measuring quality of life in PIDs, including cultural and methodological considerations, to better frame the study's focus.
3. The introduction should also present the study’s hypotheses and research questions explicitly before moving into the methodology and results. Moreover, these hypotheses and questions should be revisited in the discussion section, where the authors can compare their findings with their initial expectations and existing literature.
4. The methodology appears robust and appropriate for the research objectives, and I have no concerns about its implementation. However, the Conclusions section require rewording to make them accessible to a broader audience, including readers who may not be familiar with scientific or technical language.
5. In the bibliography of the article, only 8 out of 32 cited publications refer to studies published in the last 5 years. For such a broad and specialized study on the quality of life of people with intellectual disabilities and their physical activity, this number should be significantly higher. It is recommended that at least 50% of the cited literature come from recent years to ensure that the findings and conclusions are based on the latest data and reflect the current state of knowledge in this field.
I keep my fingers crossed for the ultimate success of the authors and the publication.
Author Response
We would like to express our gratitude to the anonymous reviewers for providing us with such valuable feedback and comments regarding the submission to European Journal of Investigation in Health, Psychology and Education entitled "Do people with intellectual disability have a better quality of life if they are physically active?”. We do appreciate reviewers’ comments and suggestions, which we have attempted to address. It is our thought that the manuscript has been significantly improved, therefore.
We have highlighted the changes in the revised version of the manuscript in a yellow-coloured text aiming to facilitate reviewers’ evaluation.
Comment 0: The manuscript is well-structured and provides an overview of the context and existing research on the relationship between physical activity and quality of life among people with intellectual disabilities. However, several important aspects are missing that would be essential for a comprehensive understanding of the subject:
Response 0: Thank you very much.
Comment 1: The authors should include in the introduction a deeper exploration of the barriers faced by PIDs in accessing physical sports activities, as well as potential strategies to overcome these barriers. This would provide valuable context for interpreting the findings.
Response 1: Thank you very much for your comment. The manuscript has been revised and completed (line 50-56)
Comment 2: The introduction would benefit from a discussion on the specific challenges of measuring quality of life in PIDs, including cultural and methodological considerations, to better frame the study's focus
Response 2: Thank you very much for your suggestion. A brief paragraph and a few references have been included to discuss such challenges (lines 64-74).
Comment 3: The introduction should also present the study’s hypotheses and research questions explicitly before moving into the methodology and results. Moreover, these hypotheses and questions should be revisited in the discussion section, where the authors can compare their findings with their initial expectations and existing literature.
Response 3: Thank you for your comment. Both the last paragraph in the introduction and the first paragraph in the discussion reflect the aim and hypothesis of the current study. As recommended by the reviewer, a specific research question has been formulated and incorporated in those lines (lines 98-105).
Comment 4: The methodology appears robust and appropriate for the research objectives, and I have no concerns about its implementation. However, the Conclusions section require rewording to make them accessible to a broader audience, including readers who may not be familiar with scientific or technical language.
Response 4: We do appreciate your positive feedback on methodology. The text in the Conclusions section has been modified aiming to make it more accessible to general audience (lines 331-344).
Comment 5: In the bibliography of the article, only 8 out of 32 cited publications refer to studies published in the last 5 years. For such a broad and specialized study on the quality of life of people with intellectual disabilities and their physical activity, this number should be significantly higher. It is recommended that at least 50% of the cited literature come from recent years to ensure that the findings and conclusions are based on the latest data and reflect the current state of knowledge in this field.
Response 5: Thank you very much for your comment. The manuscript has been revised and adjusted (lines 56, 63, 65, 111, 151, 153, 328, 344, 365, 368, 380, 387, 397, 412 and 418)
Comment 6: I keep my fingers crossed for the ultimate success of the authors and the publication.
Response 6: Thank you very much for your kind words.
Round 2
Reviewer 2 Report
Comments and Suggestions for Authors
Thank you for significantly improving the article in line with the suggestions I provided in my review. I appreciate the effort put into addressing my comments and making the necessary revisions.